# An empirically derived recommendation for the classification of body dysmorphic disorder: Findings from structural equation modeling

**Andrea Sabrina Hartmann**[1]*, **Thomas Staufenbiel**[1], **Lukas Bielefeld**[1], **Ulrike Buhlmann**[2], **Nina Heinrichs**[3], **Alexandra Martin**[4], **Viktoria Ritter**[5], **Ines Kollei**[6], **Anja Grocholewski**[7]

1 Institute of Psychology, Osnabrück University, Osnabrück, Germany, 2 Institute of Psychology, Münster University, Münster, Germany, 3 Department of Psychology, University of Bremen, Bremen, Germany, 4 Institute of Psychology, Wuppertal University, Wuppertal, Germany, 5 Institute of Psychology, Goethe University Frankfurt, Frankfurt, Germany, 6 Institute of Psychology, Otto-Friedrich-University of Bamberg, Bamberg, Germany, 7 Institute of Psychology, Technische Universität Braunschweig, Brunswick, Germany

* andrea.hartmann@uni-osnabrueck.de

**Data Availability Statement:** All relevant data are within the paper and supporting information files.

## Abstract

Body dysmorphic disorder (BDD), together with its subtype muscle dysmorphia (MD), has been relocated from the Somatoform Disorders category in the DSM-IV to the newly created Obsessive-Compulsive and Related Disorders category in the DSM-5. Both categorizations have been criticized, and an empirically derived classification of BDD is lacking. A community sample of $N = 736$ participants completed an online survey assessing different psychopathologies. Using a structural equation modeling approach, six theoretically derived models, which differed in their allocation of BDD symptoms to various factors (i.e. general psychopathology, somatoform, obsessive-compulsive and related disorders, affective, body image, and BDD model) were tested in the full sample and in a restricted sample ($n = 465$) which indicated primary concerns other than shape and weight. Furthermore, measurement invariance across gender was examined. Of the six models, only the body image model showed a good fit ($CFI = 0.972$, $RMSEA = 0.049$, $SRMR = 0.027$, $TLI = 0.959$), and yielded better $AIC$ and $BIC$ indices than the competing models. Analyses in the restricted sample replicated these findings. Analyses of measurement invariance of the body image model showed partial metric invariance across gender. The findings suggest that a body image model provides the best fit for the classification of BDD and MD. This is in line with previous studies showing strong similarities between eating disorders and BDD, including MD. Measurement invariance across gender indicates a comparable presentation and comorbid structure of BDD in males and females, which also corresponds to the equal prevalence rates of BDD across gender.

**Funding:** AG received financial support from the German Research Foundation (Deutsche Forschungsgemeinschaft; DFG; https://www.dfg. de/; GR 4761/2-1) for a Scientific Network for Body Dysmorphic Disorder. The funder had no role in study design, data collection and analysis, decision to publish, or preparation of the manuscript.

**Competing interests:** The authors have declared that no competing interests exist.

## Introduction

Body dysmorphic disorder (BDD) is characterized by excessive concerns about perceived flaws in one's appearance (e.g. a crooked nose, skin blemishes, or not being sufficiently muscular in the case of the muscle dysmorphia [MD] subtype) and associated behavioral or mental rituals to hide, improve, or control these flaws [1]. Despite being a debilitating disorder [2, 3] with high rates of suicidality [4], research interest in this area has only begun to grow in the last few years. Recently, the diagnostic entity of BDD has been reassigned: While it was classified as a subtype of hypochondriasis in the Somatoform Disorders category in the fourth version of the Diagnostic and Statistical Manual of Mental Disorders (DSM-IV), it now represents a stand-alone diagnosis within the Obsessive-Compulsive and Related Disorders (OCRD) category in the DSM-5 [5, 6]. It is also set to be replaced accordingly in the upcoming 11[th] version of the International Classification of Diseases (ICD; see online release version; [7]). While there is a broad agreement on the similarities between BDD and obsessive-compulsive disorder (OCD) [8–13], the empirical fit of the BDD diagnosis within this category has not yet been tested. Such confirmation might have important implications for both research and practice, for instance with regard to comorbidity screenings, improved clinical decision making, and information for the (further) development both of interventions and of etiological models.

The previous classification of BDD as a disorder which was "hidden" behind hypochondriasis or illness anxiety disorder within the category of Somatoform Disorders [5, 6, 14] has long been criticized [15]. While BDD and somatoform disorders such as hypochondriasis or pain disorder indeed share some symptoms targeting the body in phenomenological terms [6], they differ regarding other demographic variables (e.g. age and marital status) and clinical variables such as the focus of concern (i.e. worrying about physical health vs. worrying about appearance) [16]. Moreover, several somatoform disorders and BDD [17, 18] amongst other disorders share a heightened likelihood for dissociation and thus might be linked via dissociation as a psychopathological organizer [19, 20]. Despite widespread discussion, no study to date has tested the empirical fit of BDD within a spectrum encompassing these disorders in a so-called somatoform model.

The new classification of BDD as a stand-alone diagnosis in the category of OCRD in the DSM-5 [6] is mainly a response to criticisms of the previous version [21], and is mirrored by the proposed ICD-11 criteria [7]. It meets the expectation that disorders which are grouped together should not only present comparably on a phenomenological level (for instance intrusive thoughts and compulsive behavior in both OCD and BDD [6]), but should also show similarities in various clinical aspects, e.g. comorbidity, heredity, effective treatments, and treatment outcomes (again supported for OCD and BDD) [8–13, 22]. The relationships of BDD with the other disorders in this new DSM-5 category, namely hoarding disorder, trichotillomania, and excoriation disorder, are less strong, and the combination of these disorders under the umbrella of OCRD has been criticized [23]. Thus, while this category might be more empirically derived than the previous one, as it takes into account more diverse disorder-related aspects (see above), no study has yet tested the new classification of BDD into an OCRD spectrum by testing a so-called OCRD model with regard to its fit to the data.

In addition to the limited empirical validation of the current category, several studies have provided evidence that in terms of the function of core symptoms, comorbidity, family history, and treatment response, BDD might be more strongly linked to disorders from the Anxiety Disorder category (e.g. social anxiety disorder [SAD] and panic disorder) than to the other disorders from the OC spectrum. In particular, several reviews highlighted the similarities between and relatedness of BDD and SAD [23–26]. There is also evidence of an association

with depression across important domains including comorbidity, family history, course of the disorder, and cognitive biases [27, 28]. Thus, conceptualizing BDD as a disorder within a broader affective spectrum alongside anxieties, OCD, and depression might be another option (affective model).

Moreover, there is a fair amount of research hinting at a comparability of BDD and eating disorders. These disorders not only share the hallmark feature of a disturbed body image [29], but also show resemblances regarding onset and course as well as cognitive biases [30–32], with a particular similarity between anorexia nervosa and BDD [31]. Therefore, it has been proposed that BDD and the eating disorders might form a body image spectrum of disorders [33]. Such a categorization could also have the advantage of including muscle dysmorphia (MD), a subtype of BDD. The classification of MD has been a topic of discussion ever since it was coined as "reverse anorexia" by Pope et al. [34], given its large symptom overlap with the eating disorders [35]. Dos Santos Filho et al. [36] concluded from a systematic review that there is not sufficient scientific evidence to support the inclusion of MD in any existing category of psychological disorders. However, a conceptualization of BDD with MD as a subtype, in a body image disorders spectrum within a so-called body image model that also includes eating disorders, might pave the way for future discussions [37].

Finally, BDD still shows some unique features with respect to symptomatology, which actually complicates its identification and treatment [38]. This indicates that BDD might be seen as a unique factor which is separate from somatoform, affective, and eating disorders, as has already been suggested in an adolescent sample [39]. A major benefit of such a BDD model may lie in greater diagnostic accuracy and better treatment, by prompting health care workers to check for the unique characteristics of BDD [40].

Despite the aforementioned evidence to support the various classification suggestions, no investigation has been conducted using a bottom-up approach, with the exception of the aforementioned study in adolescents by Schneider and colleagues [39], which supported the BDD-only model. That is, no study has used real data across various symptoms and fitted these data to stipulated classification models. While Schneider et al. [39] provided valuable insights into the uniqueness of BDD symptoms in the classification in adolescents; their sample had a narrow age range (around the age of onset of BDD). Moreover, other relevant psychopathology (e.g. depressive symptoms, eating disorders; [41, 42]) has not yet been comprehensively examined. Furthermore, Schneider et al.'s study failed to include the BDD subtype MD, which may have influenced the proximity to eating disorders in terms of classification. As the study also excluded somatoform disorders, the authors were unable to provide information about the usefulness of the DSM-IV/ ICD-10 classification of BDD. Finally, the study did not include OC-related disorders other than OCD itself, such as skin picking and hair pulling, thus reducing the informative value of the OCRD model.

Therefore, the aim of the present study was to provide empirically derived recommendations for the classification of BDD and MD in adults. We thus analyzed the best classification of both BDD and MD symptoms based on shared phenomenological and comorbidity aspects, taking into account various different aspects of related psychopathology such as those of OC and related disorders (OCD, skin picking, and hair pulling), eating disorders, SAD and panic disorders, depression, illness anxiety disorder, and somatoform disorder. To assess these aspects of psychopathology, a large community and student sample completed an online survey. Based on the reviewed literature, mainly of transdiagnostic studies comparing BDD mostly with OCD, eating disorders, anxiety, and depression as well as the study by Schneider and colleagues [39] analyzing psychopathological data in adolescents, but also past and current classification of BDD, six potential classification models were specified. Models differed according to their classification of BDD symptoms within different categories and consisted of

a general psychopathology (in which one factor loaded on all indicators), somatoform, an OCRD, an affective, a body image, and a BDD model. The best fitting model was also tested with respect to equivalence in male and female subsamples due to differences in prevalence rates of disorders included as well as gender-specific subtypes of specific disorders, in particular BDD with the subtype MD. Furthermore, the models were retested in a subgroup that indicated primary concerns other than weight and shape on the questionnaire assessing BDD symptoms. This should rule out an inadvertent overestimation of the association between eating disorder and BDD pathology due to high scores on the BDD measures originally stemming from high shape and weight concerns.

## Materials and method

### Participants and recruitment

The ethics committee of Osnabrück University has approved the study (no approval number available). Written informed consent was obtained by the participants. Recruitment was conducted through university press releases at the institutions of most authors, advertisements on social media and flyers, with the aim to recruit a non-clinical community sample of participants aged 18 years and older. We chose a community-based rather than a clinical sample in order to ensure that the comorbidity structures were representative of the general population [43]. Of $N = 1166$ persons who began the survey, $n = 743$ completed it (i.e., filled out all questionnaires included in this main manuscript results of two further questionnaires will be reported elsewhere); three participants were excluded as they were below the age of 18 years and four participants were excluded because they did not clearly identify as male or female. Thus, 736 participants were included in the analyses. As reimbursement for study participation, all participants who completed the survey were given the opportunity to enter a raffle to win one of 20 online shopping vouchers worth 20 Euros each, or received student credit if they were students at one of the participating institutions.

### Procedure

The questionnaires (see description below) were programmed in the online survey software Unipark (QuestBack GmbH, Cologne Germany). The landing page of the survey informed potential participants about the study aim (assessment of various different symptoms in order to improve assessment and consequently treatment options) and duration (approximately 45 minutes) as well as privacy and confidentiality aspects of the study. After providing informed consent and completing completing the survey, participants who indicated their interest in taking part in the raffle or in receiving student credits were redirected to a separate page on which they provided their contact information.

### Measures

In the following, the employed measures are described in order of their appearance in the survey. If not otherwise indicated the total scores of the instruments were employed. Instruments were chosen based on their brevity, representativeness of symptomatology, and frequency of use. Internal consistencies of these scores are presented in Table 1 and are good to excellent.

**Demographic questionnaire.** Assessed demographic and clinical variables included gender, age, sexual orientation, educational attainment, psychotropic medication, and current or lifetime diagnosis of a mental disorder.

The German-language *Body Dysmorphic Symptoms Inventory* (Fragebogen Körperdysmorpher Symptome, FKS; [44]) consists of 18 items assessing body dysmorphic disorder

**Table 1. Univariate statistics and reliability of symptom measures.**

| Symptoms | Measure | Mean | SD | Range | k | α |
|---|---|---|---|---|---|---|
| BDD | FKS | 0.94 | 0.69 | 0..4 | 16 | .92 |
| Eating disorder | EDE-Q | 1.65 | 1.44 | 0..6 | 22 | .96 |
| Depressive | PHQ-9 | 0.87 | 0.64 | 0..3 | 9 | .89 |
| OCD | OCI-R | 0.74 | 0.57 | 0..4 | 18 | .88 |
| Health anxiety | mSHAI | 2.13 | 0.92 | 1..5 | 14 | .95 |
| Somatoform | HEALTH-49 | 0.98 | 0.71 | 0..4 | 7 | .78 |
| SAD | LSAS | 0.74 | 0.51 | 0..3 | 48 | .96 |
| MD | MDDI | 2.06 | 0.60 | 1..5 | 13 | .78 |
| Skin picking | SPS-R | 0.33 | 0.57 | 0..4 | 8 | .93 |
| Hair pulling | MGHHPS | 0.06 | 0.29 | 0..4 | 7 | .93 |
| Panic | ACQ | 1.45 | 0.45 | 1..5 | 14 | .83 |

$N = 736$. Range = minimum and maximum value of response scale. $k$ = number of items. α = Cronbach's alpha. BDD, body dysmorphic disorder; OCD, obsessive-compulsive disorder; SAD, social anxiety disorder; MD, muscle dysmorphia.

symptoms. The items can be summed into two subscales: "Specific BDD Symptoms" (items 1, 4–15) and "Associated Features" (items 16–18). Items are scaled from 0 "not at all, never, do not think about it" to 4 "very strongly so, over 5-times a week, over 8 hours a day".

The *Muscle Dysmorphia Disorder Inventory* (MDDI; [45]; German version; [46]) is a 13-item measure assessing symptoms associated with MD on three subscales (Drive for Size, Appearance Intolerance, Functional Impairment). Each item is rated on a 5-point Likert scale ranging from 1 "never" to 5 "always".

The *Eating Disorder Examination-Questionnaire* (EDE-Q; German-language version; [47]) assesses eating disorder psychopathology referring to the past 28 days. It consists of 22 items which are allocated to four subscales (Dietary Restraint, Eating Concern, Weight Concern, and Shape Concern), and scaled on a seven-point Likert scale from 0 "no day, not at all" to 6 "each day, markedly". Six additional items assess the frequencies of eating disorder behaviors, but were not used in the present study.

The *Obsessive-Compulsive Inventory-Revised* (OCI-R; German-language version; [48]) is an 18-item questionnaire assessing OCD symptom severity within the last month. Each item is rated on a 5-point Likert scale ranging from 0 "not at all" to 4 "extremely". The OCI-R comprises six subscales (Washing, Obsessing, Hoarding, Ordering, Checking, and Neutralizing).

The *Liebowitz Social Anxiety Scale* (LSAS; German-language version; [49]) assesses avoidance and fear in 24 situations that are likely to elicit social anxiety. Thirteen of the 24 items refer to performance situations and the remaining eleven items assess social interaction situations. For each of the 24 situations, the clinician derives ratings of avoidance and fear experienced by the respondent in the past week on a 4-point Likert scale. The fear scale ratings range from 0 "no fear" to 3 "severe fear" while the avoidance ratings also range from 0 to 3 (based on the percentage of time spent avoiding the particular situation from 0 "never" to 3 "usually" [67–100%]").

The *Patient Health Questionnaire* depression module (PHQ-9; German-language version; [50]) is a nine-item measure of depression severity. The items are rated on a 4-point Likert scale ranging from 0 "not at all" to 3 "nearly every day".

The *Short Health Anxiety Inventory*, modified version (mSHAI; German-language version; [51]) comprises 14 items assessing the severity of health anxiety. Items are rated on a 5-point Likert scale ranging from 1 "strongly disagree" to 5 "strongly agree".

The *Massachusetts General Hospital Hairpulling Scale* (MGHHPS; [52] unpublished German translation) assesses the severity of repetitive hair pulling. It contains seven items, which are rated on a 5-point Likert scale from 0 "no urges, none, always in control" to 4 "near constant, extreme, never able to distract".

The *Skin Picking Scale-Revised* (SPS-R; German-language version; [53]) measures skin picking disorder severity and impairment using eight items rated on a 5-point Likert scale from 0 "none" to 4 "extreme".

From the *Hamburg Modules for the Assessment of Psychosocial Health*, short version (HEALTH-49 Kurzform; Hamburger Module zur Erfassung allgemeiner Aspekte psychosozialer Gesundheit für die therapeutische Praxis; [54]), we used the subscale Physical Complaints for the present study. This subscale contains seven items asking about physical pain or complaints in the last two weeks, rated on a 5-point Likert scale from 1 "not at all" to 5 "extremely".

The German-language *Questionnaire on Body-Related Fears, Cognitions and Avoidance* (AKV; Fragebogen zu körperbezogenen Ängsten, Kognitionen und Vermeidung; [55]) measures the severity of panic disorder symptoms. It encompasses the German versions of three questionnaires: the Body Sensation Questionnaire (BSQ; [56]), the Agoraphobic Cognitions Questionnaire (ACQ; [56]) and the Mobility Inventory (MI; [57]). In the present study, only the ACQ was used. The ACQ assesses the frequency of frightening or maladaptive thoughts about the consequences of panic and anxiety. It contains 14 items rated on a 5-point Likert scale ranging from 1 "thought never occurs" to 5 "thought always occurs when I am nervous".

## Data analysis

Structural equation modeling (SEM) analyses were conducted with R v3.5.0 using the package lavaan v0.6–3 [58]. For all other analyses, we used SPSS v25 (IBM; Armonk, New York, USA). SEM analyses were performed by analyzing the covariance matrix of the symptom measures using maximum likelihood estimation. Factor intercorrelations were freely estimated in each model. As is customary, the first indicator of every factor (i.e. the uppermost indicator in Fig 1) was fixed at 1. Errors were not allowed to correlate. No post hoc modifications of the models were carried out.

As the data do not follow a multivariate normal distribution, we applied Satorra-Bentler scaled $\chi^2$-statistics (*SB-* $\chi^2$; [59]) and related robust fit indices. With a larger sample size, (scaled) $\chi^2$ becomes increasingly sensitive to very small model-data discrepancies. Therefore, we focused on additional fit indices, which are less influenced by sample size. The following indices with cutoff values following the recommendations of Hu and Bentler [60] and Schermelleh-Engel, Moosbrugger, and Müller [61] are reported: (a) comparative fit index (*CFI* $\geq$ .95 acceptable, $\geq$ .97 good), (b) Tucker-Lewis index (*TLI* $\geq$ .95 acceptable, $\geq$ .97 good), (c) root mean square error of approximation (*RMSEA* $\leq$ .08 acceptable, $\leq$ .05 good), (d) standardized root mean square residual (*SRMR* $<$ .10 acceptable, $<$ .05 good), and (e) Akaike Information Criterion, *AIC*, and Bayes Information Criterion, *BIC* (no cutoff values, smaller values indicate better fit). For RMSEA, the 90% confidence interval is also reported. If the lower bound of this confidence interval is 0, the hypothesis of "exact-fit" (i.e. the null hypothesis that RMSEA in the population is 0) is retained ($\alpha$ = 0.05). In addition, the less stringent "close-fit hypothesis" (i.e. the null hypothesis that RMSEA in the population is less than or equal to .05) is tested. A p-value (denoted as $p_{close}$) of this test greater than $\alpha$ (i.e. statistically insignificant) supports the model. In addition to the global fit measures, we present the fully standardized solution of the best-fitting model. Significance tests of parameter estimates are based on robust standard errors (i.e. the MLM estimator in lavaan). These analyses were

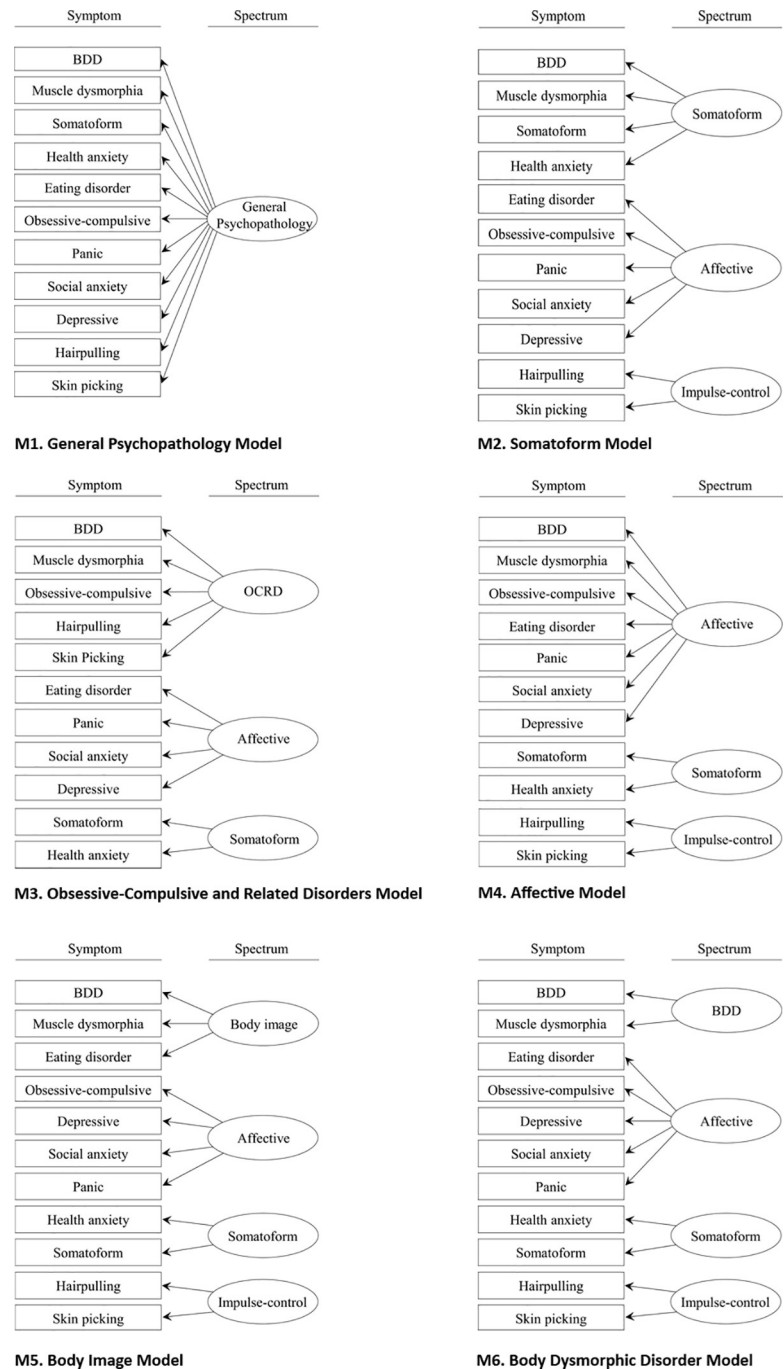

**Fig 1. Six classification models for symptoms.**

repeated with a restricted sample ($n = 465$) which did not indicate primary shape and weight concerns in the FKS (scores $\leq 2$ [scaled 0 "not at all" to 4 "very strongly so"] on item 3 "Is your main concern about your appearance that you are not thin enough or could become fat?").

If the best-fitting model shows an acceptable fit and the estimated parameters are as expected, the equivalence of the model in the male and female subsamples can be tested using multi-group SEM. In a first step, the model is tested for men and women separately. If the

model fits well in both subsamples, one proceeds with a series of hierarchically nested models which allow increasingly stronger forms of invariance to be examined [62].

The first multi-group model is the baseline model, in which all parameters in both groups are freely estimated. A good fit of this baseline model speaks in favor of configural invariance (Model CI). In this weakest form of invariance, the number of factors and their correspondence to the indicators is the same across both groups. If configural invariance has been established, metric invariance (Model MI) can be examined. In this step, all factor loadings are constrained to be equal in both groups. Compared to the baseline model, this nested restricted model should not be statistically worse, i.e. the $\Delta SB\text{-}\chi^2$ test [63] should be non-significant. Furthermore, global fit should not practically decline too much. Following Chen [64], the cutoffs for a practically significant decrease in fit employed here are $\Delta CFI \leq -0.01$, supplemented by $\Delta RMSEA \geq 0.015$, or $\Delta SRMR \geq 0.03$ (where the fit index of the restricted model is always subtracted from the fit index of the unrestricted model). If metric invariance is supported, all indicators contribute to their factor with similar magnitude and one can proceed to test for scalar invariance (Model SI). For this purpose, the intercepts are additionally constrained to be equal across the two groups. Again, the restricted model (Model SI) is then compared to the less restricted model (Model MI) and the cutoff values proposed by Chen [64] are applied, which are identical to those above with the exception of $\Delta SRMR$, which requires $\Delta SRMR \geq 0.01$. If scalar invariance can be established, it is possible to compare means on the latent factors between men and women.

## Results

### Participants' demographic

Our total sample consisted of 736 participants. Of those, 153 (20.8%; $M_{age}$ = 29.01 years, $SD_{age}$ = 11.64; $Min_{age}$ = 18, $Max_{age}$ = 61) were male and 583 (79.2%; $M_{age}$ = 24.92, $SD_{age}$ = 8.18; $Min_{age}$ = 18, $Max_{age}$ = 73) were female.

### Model fits

For all symptom scales, the respective items were averaged over non-missing data. None of the symptom scales had more than three missing values (<0.03%), with only 34 missing values across all items and scales in total (<0.03%). Table 1 displays means, standard deviations, and reliability information of the symptom scales. The bivariate product moment correlations $r$ between the symptom scales are presented in Table 2. Since data were not bivariate normally distributed we employed bootstrapping instead of the standard parametric significance tests for r (against 0, percentile method, 1000 samples).

Global SEM fit indices of the six classification models (as presented in Fig 1) are shown in Table 3. The results reveal a very clear picture, with the 4-factor Body Image model (BIM) outperforming the other competing models. All global fit indices of the BIM indicated a good fit (*CFI* = 0.972, *RMSEA* = 0.049, *SRMR* = 0.027) or an acceptable fit (*TLI* = 0.959). According to the cutoffs reported above, none of the competing models yielded an acceptable fit on any of the fit measures with the exception of *SRMR*. Furthermore, the BIM showed a statistically better fit to the data than the general psychopathology model, $\Delta SB\text{-}\chi^2(6) = 277.24$, $p < .001$. Since they are not nested, the BIM and the remaining models could only be compared by means of *AIC* and *BIC*. The BIM produced the lowest *AIC* and *BIC* values of all models, again indicating the best relative fit.

The BIM with completely standardized parameter estimates is shown in Fig 2. As expected, all factor loadings were positive and statistically significant ($p < .01$). The smallest loading was exhibited by the symptom "Hair pulling" on the factor "Impulse-control". All latent intercorrelations between the four factors were positive and statistically significant ($p < .01$). Notably,

**Table 2. Pearson correlations between symptom measures.**

|  | FKS | EDE-Q | PHQ-9 | OCI-R | mSHAI | HEALTH-49 | LSAS | MDDI | SPS-R | MGHHPS | ACQ |
|---|---|---|---|---|---|---|---|---|---|---|---|
| FKS |  | .540* | .390* | .346* | .363* | .311* | .373* | .521* | .274* | -.002 | .350* |
| EDE-Q | .798* |  | .379* | .278* | .309* | .323* | .346* | .542* | .226* | .041 | .333* |
| PHQ-9 | .538* | .536* |  | .489* | .404* | .557* | .572* | .353* | .244* | .127* | .518* |
| OCI-R | .373* | .344* | .484* |  | .406* | .387* | .457* | .280* | .209* | .012 | .399* |
| mSHAI | .383* | .340* | .413* | .428* |  | .480* | .377* | .268* | .236* | .043 | .414* |
| HEALTH-49 | .425* | .405* | .621* | .391* | .457* |  | .431* | .285* | .198* | .076 | .529* |
| LSAS | .480* | .446* | .600* | .438* | .343* | .427* |  | .362* | .228* | .065 | .556* |
| MDDI | .676* | .684* | .451* | .325* | .307* | .394* | .426* |  | .149* | -.012 | .275* |
| SPS-R | .271* | .202* | .303* | .239* | .188* | .262* | .267* | .177* |  | .187* | .255* |
| MGHHPS | .073* | .088* | .123* | .045 | .026 | .077* | .079* | .102* | .152* |  | .065 |
| ACQ | .453* | .402* | .556* | .422* | .455* | .562* | .554* | .387* | .252* | .087* |  |

Correlations from full sample (N = 736) below the diagonal and from restricted sample (N = 465) above the diagonal.

* p < .05 (two-tailed) applying bootstrapping. FKS, Body Dysmorphic Symptoms Inventory (Fragebogen Körperdysmorphe Störung); EDE-Q, Eating Disorder Examination-Questionnaire; PHQ-9, Patient Health Questionnaire depression module; OCI-R, Obsessive-Compulsive Inventory-Revised; mSHAI, Short Health Anxiety Inventory, modified version; HEALTH-49, Hamburg Modules for the Assessment of Psychosocial Health, short version; LSAS, Liebowitz Social Anxiety Scale; MDDI, Muscle Dysmorphia Disorder Inventory; SPS-R, Skin Picking Scale-Revised; MGHHPS, Massachusetts General Hospital Hairpulling Scale; ACQ, Agoraphobic Cognitions Questionnaire.

the correlation between the factors "Affective" and "Somatoform" was very high ($\rho = 0.93$, $p < .001$).

## Gender invariance of the body image model

Building on the preceding results, the invariance analyses refer to the best-fitting BIM. Testing this model for both groups separately revealed good to acceptable fit for men and women (see Table 4). With one exception, all factor loadings were higher than 0.56 (completely standardized loadings, $\lambda^*$) and statistically significant, $p < .001$. The exception was the loading of the symptom "Hair pulling" on the factor "Impulse-control" (women: $\lambda^* = 0.20$, $p < .05$; men: $\lambda^* = 0.45$, $p < .01$), which was also the lowest-loading symptom in the total sample.

In the next step, parameters in both groups were estimated simultaneously. This baseline model yielded an acceptable fit, which speaks in favor of configural invariance and justifies the evaluation of more restrictive invariance models. The constraint of equal factor loadings in both groups produced a statistically significant increase in misfit (Model MI compared to

**Table 3. SEM fit statistics for six competing models in the complete sample.**

| Model | SB-$\chi^2$ | df | CFI | TLI | RMSEA [90% CI] | $p_{close}$ | SRMR | AIC | BIC |
|---|---|---|---|---|---|---|---|---|---|
| 1 Gen.Patho | 500.695 *** | 44 | 0.805 | 0.757 | 0.119 [0.111, 0.127] | .00 | 0.068 | 12,543.647 | 12,644.874 |
| 2 Somato. | 501.266 *** | 41 | 0.804 | 0.737 | 0.124 [0.115, 0.132] | .00 | 0.068 | 12,534.983 | 12,650.014 |
| 3 OCRD | 457.312 *** | 41 | 0.822 | 0.762 | 0.117 [0.109, 0.126] | .00 | 0.071 | 12,483.153 | 12,598.183 |
| 4 Affective | 486.446 *** | 41 | 0.810 | 0.745 | 0.121 [0.113, 0.130] | .00 | 0.069 | 12,509.634 | 12,624.665 |
| 5 BIM | 104.096 *** | 38 | 0.972 | 0.959 | 0.049 [0.039, 0.059] | .57 | 0.027 | 12,007.593 | 12,136.427 |
| 6 BDD | 413.851 *** | 38 | 0.840 | 0.768 | 0.116 [0.107, 0.125] | .00 | 0.059 | 12,411.997 | 12,540.832 |

N = 736. Gen.Patho., General Psychopathology Model; Somato., Somatoform Model; OCRD, Obsessive-Compulsive and Related Disorders Model; Affective, Affective Model; BIM, Body Image Model; BDD, Body Dysmorphic Disorder Model.

*** p < .001

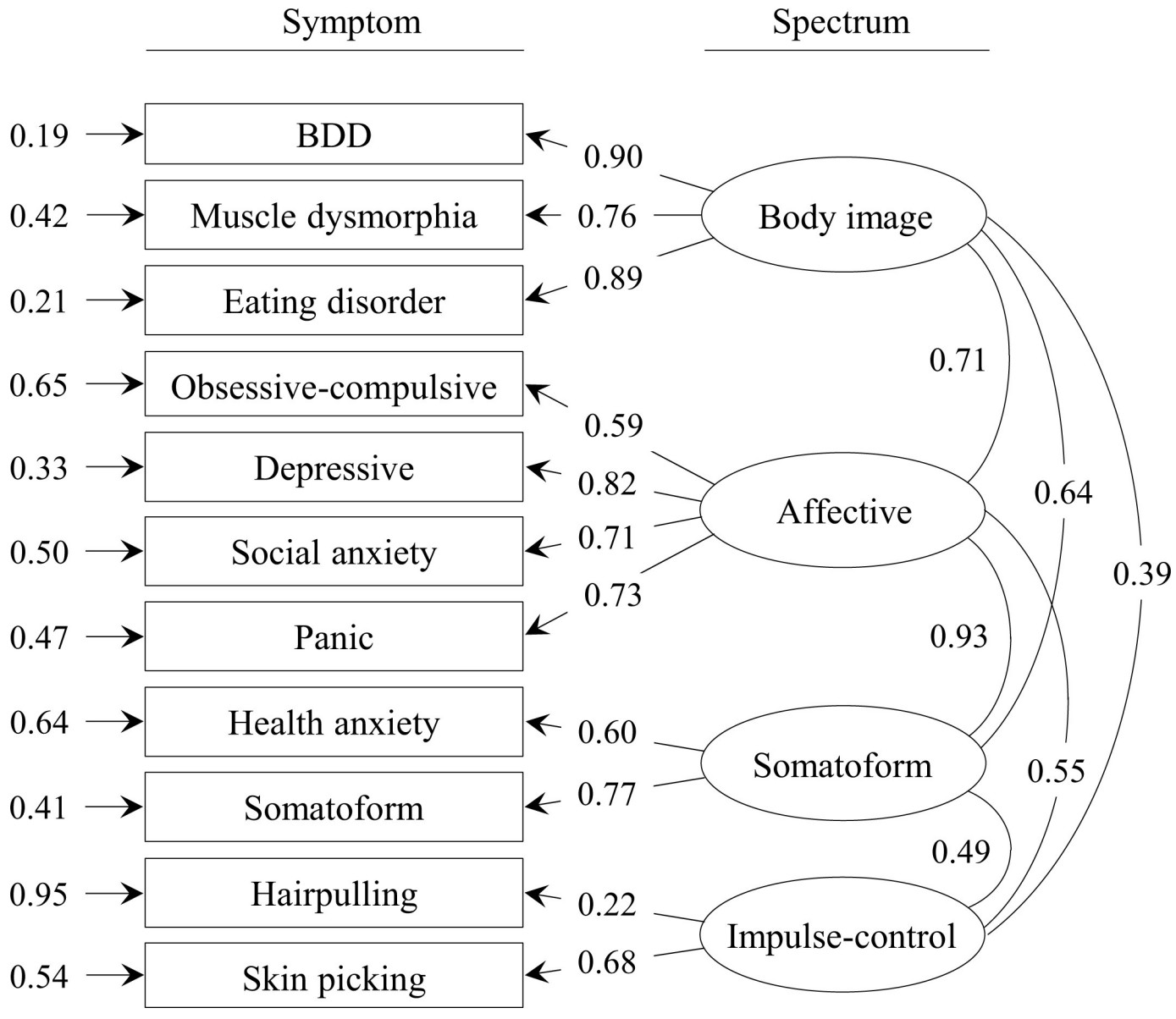

**Fig 2. Completely standardized parameter estimates for the body image model.**

Model CI, see Table 5). This speaks against metric invariance. On the other hand, the decrease in fit of the other indices was small and below the cutoff values proposed by Chen [64]. When examining the freely estimated loadings of the baseline model in detail, it became evident that in particular, the estimate of the path from the factor "Affective syndrome" to the "Obsessive-compulsive" symptoms diverged in the two subsamples. Therefore, we tested a model for partial metric invariance (Model PMI) with this path being freely estimated in both groups [65]. Freeing this loading resulted in a stronger standardized path for men ($\lambda^* = 0.710$) than for women ($\lambda^* = 0.574$). Compared to the baseline model, no statistically and no practically significant increase in misfit occurred.

As we obtained at least partial support for metric invariance, we also tested for scalar invariance (Model SI). Using the metric and the partial metric invariance model as bases, we

**Table 4. Model fit for invariance models.**

| Model | Invariance | SB-$\chi^2$ | df | CFI | TLI | RMSEA [90% CI] | SRMR |
|---|---|---|---|---|---|---|---|
| M | - | 48.381 | 38 | 0.971 | 0.958 | 0.042 [0.000, 0.071] | 0.045 |
| F | - | 109.961*** | 38 | 0.964 | 0.948 | 0.057 [0.046, 0.068] | 0.033 |
| CI | configural | 158.152*** | 76 | 0.963 | 0.946 | 0.054 [0.044, 0.065] | 0.033 |
| MI | metric | 174.555*** | 83 | 0.958 | 0.945 | 0.055 [0.045, 0.065] | 0.044 |
| PMI | partial metric | 162.626*** | 82 | 0.963 | 0.951 | 0.052 [0.041, 0.062] | 0.037 |
| SI | scalar | 203.365*** | 90 | 0.948 | 0.937 | 0.059 [0.049, 0.068] | 0.045 |
| PSI | partial scalar | 197.082*** | 89 | 0.951 | 0.939 | 0.057 [0.048, 0.067] | 0.041 |

$N = 736$ for all models except F and M. $N_F = 583$ (female), $N_M = 153$ (male).

*** $p < .001$

constrained the intercepts to be equal across both groups. For both constrained models (SI and PSI, see Table 5), the scaled *SB-$\chi^2$* difference tests indicated significant increases in misfit. For model PSI, Δ*CFI* slightly exceeded the cutoff value. The change in the other model fit indices for models SI and PSI remained below the cutoffs. Although the results are somewhat inconsistent, there was not sufficient evidence to assume scalar invariance of the BIM in male and female subsamples. Thus, the BIM model demonstrated configural invariance and partial metric invariance. This means that the basic correspondence of symptoms to the underlying four factors is similar in men and women.

## Replication of model fits in a sample without primary shape and weight concerns

For the restricted sample of participants who indicated that their main appearance concern did not refer to weight and/or shape, the correlations between the symptom scales are displayed above the diagonal in Table 2. We compared the six competing models in the same way as for the complete sample. From Table 6, it is apparent that the results of the SEM analyses showed the same pattern. The BIM fitted the data well, and better than all other models. Invariance analyses were not replicated, since the sample size of men in the restricted sample was only $n = 112$. Furthermore, for some models, improper SEM solutions emerged.

## Discussion

The present study sought to provide an empirically based recommendation for a future classification of body dysmorphic disorder. To this aim, different classification models, i.e. with BDD symptoms being allocated to different diagnostic categories (somatoform, OCRD, affective, body image and BDD model), were tested using a structural equation modeling approach

**Table 5. Model comparisons for invariance models.**

| Model | Invariance | ΔSB-$\chi^2$ | Δdf | *p* | ΔCFI | ΔRMSEA | ΔSRMR |
|---|---|---|---|---|---|---|---|
| MI | metric | 16.231 | 7 | 0.023 | −0.005 | 0.001 | 0.011 |
| PMI | partial metric | 5.094 | 6 | 0.532 | 0.000 | −0.002 | 0.004 |
| SI | scalar | 37.191 | 7 | 0.000 | −0.010 | 0.004 | 0.001 |
| PSI | partial scalar | 47.056 | 7 | 0.000 | −0.012 | 0.005 | 0.004 |

$N = 736$. $p = p$-value of ΔSB-$\chi^2$ test. Models MI and PMI are compared to model CI, model SI is compared to model MI, and model PSI is compared to model PMI.

**Table 6. SEM fit statistics for six competing models, restricted sample due to exclusion of primary shape and weight concerns.**

| Model | SB-$\chi^2$ | df | CFI | TLI | RMSEA [90% CI] | $p_{close}$ | SRMR | AIC | BIC |
|---|---|---|---|---|---|---|---|---|---|
| 1 Gen.Patho | 188.082*** | 44 | 0.852 | 0.814 | 0.084 [0.073, 0.095] | .00 | 0.064 | 6,658.528 | 6.749.653 |
| 2 Somato. | 186.069*** | 41 | 0.851 | 0.799 | 0.087 [0.076, 0.099] | .00 | 0.061 | 6,649.603 | 6,753.154 |
| 3 OCRD | 173.078*** | 41 | 0.864 | 0.817 | 0.083 [0.072, 0.094] | .00 | 0.063 | 6,643.321 | 6,746.872 |
| 4 Affective | 180.511*** | 41 | 0.856 | 0.807 | 0.086 [0.074, 0.097] | .00 | 0.060 | 6,639.713 | 6,743.263 |
| 5 BIM | 54.462 * | 38 | 0.983 | 0.975 | 0.031 [0.011, 0.046] | .98 | 0.029 | 6.483.638 | 6,599.616 |
| 6 BDD | 140.840*** | 38 | 0.894 | 0.847 | 0.076 [0.064, 0.088] | .00 | 0.055 | 6,592.086 | 6,708.063 |

N = 465. Gen.Patho., General Psychopathology Model; Somato., Somatoform Model; OCRD, Obsessive-Compulsive and Related Disorders Model; Affective, Affective Model; BIM, Body Image Model; BDD, Body Dysmorphic Disorder Model

* $p < .05$

*** $p < .001$

in a community-based sample of adults. Additionally, gender differences in model fits were examined.

Interestingly, neither the model that conceptualized BDD symptoms within an OCRD category nor the model with BDD symptoms as a stand-alone factor showed a satisfactory fit to the data. For the OCRD model, this strongly contradicts the current classification of BDD as a disorder in the DSM-5 [6] and the proposed classification in the ICD-11 [7] as well as studies highlighting similarities between OCD and BDD [66]. However, other empirical studies have also revealed large differences in phenomenology as well as in clinical and personality characteristics between BDD and other diagnoses of the category such as skin picking, hair pulling, or hoarding, and their combination within one category has received a great deal of criticism [23]. With regard to the model that proposes BDD symptoms to be a stand-alone factor, the findings of the present study are in contrast to those of Schneider and colleagues [39], who found this to be the best-fitting model. Schneider and colleagues [39] argued that there might be symptoms that stand out as unique in BDD in adolescents, e.g. the extent of delusionality with which beliefs are held, which are not mirrored in other disorders. While the same symptoms might still be unique in adulthood, further comorbidities might have led to a more inclusive picture of phenomenology. This assumption is supported by the finding that comorbid diagnoses (e.g. depression, SAD, OCD, and certain eating disorders such as bulimia nervosa) most often develop only after the onset of BDD [67–71]. Thus, while the main symptoms do not seem to differ between adolescents and adults [72], comorbidity patterns might change, impacting the recommendations for an ideal classification.

Furthermore, both models including BDD symptoms in an affective or a somatoform model did not show an adequate fit, despite previous evidence of similarities between BDD and anxiety disorders (particularly SAD) and depression, as well as high comorbidity rates [28, 73]. At first glance, this is in contrast to previous findings which confirmed higher-order dimensions consisting of a whole range of internalizing (primarily emotional) disorders and externalizing syndromes, as proposed and first identified by Krueger in youth [74] and in adults [75]. However, a later study showed that this model is not developmentally stable and robust against the addition of further disorders, including eating disorders and somatoform disorders [76], which supports the current findings. Regarding the somatoform model, our findings are in accordance with empirical evidence showing little similarity between BDD and other disorders in the category, despite sharing a focus on the body [15]. Moreover, they further support the move of the diagnosis out of the DSM-IV [5] category Somatoform Disorders with the introduction of the DSM-5 [6].

Surprisingly, the model in which BDD symptoms are part of a body image disorders spectrum showed the best fit. This finding remained when the sample was reduced to those participants who indicated primary concerns other than shape and weight in the FKS. These participants were analyzed separately in order to prevent an overestimation of the correlation between ED and BDD symptom levels due to shared symptoms such a checking, avoidance, and behaviors to change the body such as dieting. This corroborates previous findings on the comparability of EDs and BDD on a whole range of clinical, personality, and cognitive variables [30]. Regarding non-clinical samples, Samad and colleagues [77] found that while ED and BDD symptoms appear to track together, they seem to identify different sets of psychopathological features. This supports our results that the two symptomatologies might best be captured under one category. Furthermore, such a classification would also accommodate the discussion around the BDD subtype MD, which was conceptualized as a variant of eating disorders in earlier research [33] and which shows great phenomenological similarities with eating disorders (e.g. dieting [78]). As a consequence, several authors have suggested a body image model incorporating the eating disorders with a clear reference to body image (e.g. anorexia and bulimia nervosa and potentially binge eating disorder) together with BDD and MD (e.g. [36, 79]). Disorders falling in such a category would be characterized by a body image disturbance, involving an affective component such as body dissatisfaction, a misperception of the body, cognitive distortions such as attention and interpretation biases, and behaviors such as checking, avoidance, and appearance fixing [80, 81]. Thus, these disorders might present with symptoms also present in other disorders such as sadness or OCD-like checking, but they are all exclusively related to the body. Disorders within this category might be most easily differentiated by examining the body parts individuals are most dissatisfied with, i.e., body zones typically linked to weight concerns in EDs, particularly anorexia nervosa, whereas facial features, hair and skin in BDD [82].

The body image model also achieved partial metric invariance, but not scalar invariance, across gender when the path from the factor "affective syndrome" to "OCD symptoms" was freed. The findings suggest that "affective syndrome" influences OCD symptoms more strongly in men than in women. This is in line with the analysis of the comorbidity structure of OCD in the two genders, which hints at a stronger comorbid presentation with affective disorders in men, and a stronger association of the religious obsessive-compulsive dimension (which is more typical in men) with depression (e.g. [83]). In sum, the basic correspondence of symptoms to the underlying four factors is similar in men and women. While this is in contrast to the findings of comparable studies for eating disorders (e.g. [84]), these findings might be due to the more equally distributed prevalence of BDD across gender compared to EDs (e.g. [4, 85]). Furthermore, this finding is also in contrast to the study by Schneider and colleagues [39], who did not find metric invariance for their best-fitting model with the stand-alone BDD factor in adolescents. Developmental tasks in adolescence (e.g. findings one's own gender identity; [86]) might be more gender-specific than tasks in early or middle adulthood, thus leading to gender-specific differences in the development of comorbidities.

The present findings need to be interpreted in the light of the strengths and limitations of the study. Limitations pertaining to the sample might lie in the recruitment of a community-based, non-clinical sample, as in particular, the structures of pathology and comorbidity may differ in clinical samples [87]. Along the same lines, we have only examined BDD symptoms independent of those warranting a diagnosis of BDD. Therefore, it might be advisable to replicate these findings in a clinical sample with subthreshold levels of symptoms looking into the comorbidity structure at the disorder level. Furthermore, participants were not recruited at random, which might limit generalizability of findings. Additionally, the sizes of the female and male samples differed substantially, meaning that gender differences between groups should be interpreted with caution. Moreover, the high level of education of the sample limits

the ability to generalize the findings to persons with a more diverse educational background. Finally, from a design point of view, while the data-driven bottom-up approach of the present study can be criticized, it did allow for the identification of symptom clusters and the examination of empirically derived models. Also, as can be seen from Fig 1, some factors were assessed only by few indicator variables. In general, simulations show that more indicators per factor result in fewer nonconverged solutions, fewer improper solutions and more accurate parameter estimates (MBR; [88]). But simulations demonstrate as well that a low number of indicators can be compensated by larger sample size, e.g. $N \geq 400$ [61]. Moreover, there were no signs of nonconvergent or improper solutions in any of our models. Taken together, we found no evidence that the relatively low number of indicators has caused problems. The strengths of the study include the large sample size, the broad age range and the use of validated self-report instruments covering a broad spectrum of psychopathology.

This is the first study to examine the empirical evidence for different classification models of BDD in adults, revealing a clear superiority of a body image model of BDD, MD, and eating disorders. Despite this highly important finding, it is of note, that besides this rather categorical approach of classification, recently, a dimensional one has been proposed and highly researched. Along the lines of the initiative Research Domain Criteria (RDoC) of the National Institute of Mental Health (e.g., [89]), researchers aim to organize mental disorders not only based on symptoms and behaviors presented but rather along domains of human functioning such as negative valence or cognition that can be measured rather diversely (e.g., self-report vs. on a physiological or a genetic level). Thus, future research might also want to be focused on finding common psychopathological organizers of BDD with other disorders and examine where BDD lies on these dimensions measured with different instruments, respectively.

Our findings have several clinical and research implications. From a clinical perspective, the results highlight the relatedness of EDs and BDD. Although BDD and eating disorders are also often comorbid [90–92], BDD might still go unnoticed in practice because eating disorders, particularly anorexia nervosa, might be more evident at first glance. As such, the present findings might be useful for clinicians and might lead to an improvement in the under-diagnosis of this severe mental disorder [93]. The better fit of a body image model of eating disorders may suggest that treatment techniques which bring about change in EDs might also be used in BDD. This transfer may lead to promising new avenues to tackle BDD. One such technique (e.g. mirror exposure) has already been successfully applied in BDD (e.g. [94]). In the next step, improvement of treatments could be informed by treatment of the respective other disorder. Furthermore, discussions on the development of the next generation of classification systems should take the present findings into account. With regard to non-clinical samples, the high correlation of symptoms in the present sample highlights the need to address the potentially present other symptom cluster in prevention programs. In terms of research implications, the findings need to be replicated in a clinical sample of patients with BDD. Finally, our findings should foster research comparing BDD with other disorders, in particular eating disorders, in order to establish a clear picture of similarities and differences.

## Supporting information

**S1 File. An empirically derived recommendation for the classification of body dysmorphic disorder: Findings from structural equation modeling.**
(PDF)

**S1 Data.**
(SAV)

## Author Contributions

**Conceptualization:** Andrea Sabrina Hartmann, Ulrike Buhlmann, Nina Heinrichs, Alexandra Martin, Viktoria Ritter, Ines Kollei, Anja Grocholewski.

**Data curation:** Andrea Sabrina Hartmann, Thomas Staufenbiel.

**Formal analysis:** Thomas Staufenbiel.

**Funding acquisition:** Anja Grocholewski.

**Investigation:** Lukas Bielefeld.

**Methodology:** Andrea Sabrina Hartmann, Ulrike Buhlmann, Nina Heinrichs, Alexandra Martin, Viktoria Ritter, Ines Kollei, Anja Grocholewski.

**Project administration:** Andrea Sabrina Hartmann.

**Supervision:** Andrea Sabrina Hartmann.

**Writing – original draft:** Andrea Sabrina Hartmann, Lukas Bielefeld.

**Writing – review & editing:** Thomas Staufenbiel, Ulrike Buhlmann, Nina Heinrichs, Alexandra Martin, Viktoria Ritter, Ines Kollei, Anja Grocholewski.

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
