## [Decision Letter · Decision Letter 0]

18 Feb 2020

PONE-D-19-15829

An empirically derived recommendation for the classification of body dysmorphic disorder: Findings from structural equation modeling

PLOS ONE

Dear Andrea Hartmann,

Thank you for submitting your manuscript to PLOS ONE. After careful consideration, we feel that it has merit but does not fully meet PLOS ONE’s publication criteria as it currently stands. Therefore, we invite you to submit a revised version of the manuscript that addresses the points raised during the review process.

The comments and suggestions issued by the reviewers can be seen below.

We would appreciate receiving your revised manuscript by March 20. To enhance the reproducibility of your results, we recommend that if applicable you deposit your laboratory protocols in protocols.io, where a protocol can be assigned its own identifier (DOI) such that it can be cited independently in the future. For instructions see: http://journals.plos.org/plosone/s/submission-guidelines#loc-laboratory-protocols

We look forward to receiving your revised manuscript.

Kind regards,

Flávia L. Osório, PhD

Academic Editor

PLOS ONE

Journal Requirements:

Reviewers' comments:

**Comments to the Author**

1. Is the manuscript technically sound, and do the data support the conclusions?

Reviewer #1: Yes

Reviewer #2: Yes

Reviewer #3: Partly

2. Has the statistical analysis been performed appropriately and rigorously? 

Reviewer #1: Yes

Reviewer #2: Yes

Reviewer #3: Yes

3. Have the authors made all data underlying the findings in their manuscript fully available?

Reviewer #1: No

Reviewer #2: Yes

Reviewer #3: Yes

4. Is the manuscript presented in an intelligible fashion and written in standard English?

Reviewer #1: Yes

Reviewer #2: Yes

Reviewer #3: Yes

5. Review Comments to the Author

Reviewer #1: The article describes a procedure used to recommend a classification for body dysmorphic disorder. The article is about a relevant topic, it is well-written and a pleasant reading. It considers appropriate statistical methodologies, which are well described. I have a few questions about the study.

Minor Compulsory Revisions:

1. Page 7/lines 11-13: Authors described the recruitment process. Since the sample is not at random (even at the institutions of the authors), it would be interesting to discuss the generalization of the results. At page 22 (lines 14-20), the authors mentioned some limitations of the same, but this could be further extended.

2. Page 7/ line 12: The participants are 18 years and older. At page 13/lines 6-7: the authors described the mean and standard deviation for age of the participants of the study. It might be important to include the highest age for a better description of the sample, which is cited to have a “broad age range” (page 22/line 24).

3. Page 11/line 18: What is the definition of pclose?

4. Page 13/line 13: (Pearson) bivariate product moment correlations are presented in Table 2. In the footnote of Table 2 (Page 15), statistically significant correlation coefficients are highlighted. As data do not follow a multivariate normal distribution (as mentioned at page 11/line 8), what is the adopted procedure to test the correlations?

Reviewer #2: PONE-D-19-15829 An empirically derived recommendation for the classification of body dysmorphic disorder: Findings from structural equation modeling

It is a very interesting work, very well justified, analyzed and explained. The sample used is large, the instruments used are relevant, the statistical applied are flawless, and the contributions are adequately discussed, including the limitations of the study.

The researchers are asked some questions that we consider important from a conceptual point of view. This has to be taken into account, since it is the main direction and contribution of this research.

First of all, one should not speak of BDD but of BDD symptomatology. This should be taken into account because some questionnaires or scales have been administered, but it cannot be confirmed in this way that we are talking about a formal diagnosis. The authors allude precisely to the DSM and ICE, so it is understood that we should talk about a finding of inclusion, exclusion, and impairment criteria identified through a formal diagnostic interview. In this way we could say, strictly speaking, that we are talking about BDD.

In relation to the above, the authors are suggested to refer to symptoms of BDD, or sub-threshold BDD or, as expressly referred to by the DSM, dysmorphic concerns. In the same vein, authors are recommended to review a publication that addresses precisely this issue (doi: 10.1002 / ijop.12646).

Secondly, being in full agreement with the authors about the approach adopted, and the results indicated, it is perhaps missing what aspects of body image are relevant here. It may seem tautological that there are disorders where the body and discomfort about it are categorized more precisely as body image disorders. It would be relevant to propose what aspects should lead the investigation if, as proposed, the validity of the construct has more to do with body image than with emotional, obsessive-compulsive symptoms, etc.

Third, in connection with the above, it is true, as the authors say, that there is no research about the relationship between somatization and BDD. It may be necessary to point out components of somatization that must be taken into account in order to really discard this relationship. Somatization cannot be analyzed simply from a list of physical complaints. The description of the absorption states that can be seen in both the OCD and the BDD is classic. Precisely for that reason, it is a less analyzed, but proposed psychopathological organizer. It is suggested to review: doi: 10.1016 / j.cpr.2004.08.006 as well as Farina, B., & Liotti, G. (2013). Does a dissociative psychopathological dimension exist? A review on dissociative processes and symptoms in developmental trauma spectrum disorders. Clinical Neuropsychiatry, 10(1), 11–18.

In short, the authors are asked to indicate which psychopathological organizers would conform to the BDD as a body image disorder, close, but differentiable from the Eating Disorders, for example. This contribution could lead the investigation precisely on the mediating and moderating components, and not exclusively on the description of symptoms and behaviors.

Reviewer #3: The current manuscript employs structural equation modeling (SEM) to identify an empirically supported classification structure for body dysmorphic disorder (BDD). The authors generated six a priori models based on the literature and determined that a Body Image Model provided the best fit structure. This is an important and understudied area of research. Particular strengths of this study, relative to past research on this topic, are the use of SEM with strong theoretically-derived models, large sample size, and a set of well-validated indicator measures. However, there are some concerns with the analyses and suggestions described in more detail below to improve interpretability of the findings.

The authors did a commendable job in their description, justification, and interpretation of all fit indices, and examination of gender invariance and multi-group solutions. However, the main concern with the approach is with variable selection.

There would be an expected level of multicollinearity among indicator variables across the spectrum of psychopathology included in this study, but there is quite substantial collinearity between the FKS and EDE-Q, in particular, at the zero-order level (r = .798), which is only potentially problematic in that suggests there may be redundant information contributing to the “Body Image” factor. Given that this model provided the best fit, I wonder if the authors inspected reasons for this particularly high degree of overlap and considered dropping each of the variables (e.g., EDE-Q) to test its effect on parameter estimates and model fit. Were there other eating disorder inventories/scales included in the online survey? Alternatively, I wonder if there are subscales within the FKS and EDE-Q that particularly hang together driving the overall multicollinearity, which could be examined in further sensitivity analyses to defend the robustness of the Body Image model.

There were few indicator variables (as few as 2) for some of the factors. Please discuss the implications of this on parameter estimation and model comparison.

Another concern is regarding the significant amount of missing data, which is common to survey studies. How were “complete” surveys defined, and were there any attempts made to examine patterns of missingness based on available data collected?

It was clear to me that the six models tested in the current study were derived based on the existing literature; however, to an unfamiliar reader, this may have been unclear. There could be more background on the selection of measures and models included in the study. I also couldn’t find figure captions for Figures 1 and 2.

Results on clinical characteristics appear to be missing on p.13.

As the study objective was to examine the structure of BDD, it presents a challenge interpreting data based on a community (non-clinical) sample. The defining feature of BDD is that the physical features are dysmorphic, by nature, which cannot be reliably assessed via self-report. Although the authors already note these limitations in the Discussion, some of the findings are still overstated and I suggest scaling back the language accordingly. In particular, I suggest paying close attention to discussion of the findings with reference to studies conducted in large community samples that estimate the whole continuum of body dysmorphic concerns (and other related pathology). For example, the authors discuss the implications of support for a body image spectrum of classification for BDD but only in the context of clinical studies of BDD, eating disorders, and MD (p.21), rather than in non-clinical, analogue, or community samples.

6. PLOS authors have the option to publish the peer review history of their article (what does this mean?). If published, this will include your full peer review and any attached files.

Reviewer #1: No

Reviewer #2: No

Reviewer #3: No

---

## [Author Response · Author response to Decision Letter 0]

7 Apr 2020

Dear Reviewers,

Thank you very much for your valuable comments on our manuscript. We have addressed each one of them below and have performed respective changes in the manuscript which have been highlighted in yellow. In the attached coverletter, you can find the response to the Reviewers in a more formatted version also.

Reviewer #1: 

The article describes a procedure used to recommend a classification for body dysmorphic disorder. The article is about a relevant topic, it is well-written and a pleasant reading. It considers appropriate statistical methodologies, which are well described. I have a few questions about the study.

Thank you very much for this favorable evaluation of our study. We have addressed your comments below.

Minor Compulsory Revisions:

1. Page 7/lines 11-13: Authors described the recruitment process. Since the sample is not at random (even at the institutions of the authors), it would be interesting to discuss the generalization of the results. At page 22 (lines 14-20), the authors mentioned some limitations of the same, but this could be further extended.

This is an important point. We have, therefore, mentioned the missing randomness in recruitment and the resulting limited generalizability of the findings in the discussion section on page 24: 

“Furthermore, participants were not recruited at random, which might limit generalizability of findings.”

2. Page 7/ line 12: The participants are 18 years and older. At page 13/lines 6-7: the authors described the mean and standard deviation for age of the participants of the study. It might be important to include the highest age for a better description of the sample, which is cited to have a “broad age range” (page 22/line 24).

We have supplemented this important information in the text on page 13: 

“Our total sample consisted of 736 participants. Of those, 153 (20.8%; Mage = 29.01 years, SDage = 11.64; Minage=18, Maxage=61) were male and 583 (79.2%; Mage = 24.92, SDage = 8.18; Minage=18, Maxage=73) were female.” 

3. Page 11/line 18: What is the definition of pclose?

Thank you for this chance to clarify the meaning of pclose. pclose is the p-value of the test of the null-hypothesis that RMSEA in the population is ≤ .05. We have now made this clearer by rewriting the relevant section on page 11:

“For RMSEA, the 90% confidence interval is also reported. If the lower bound of this confidence interval is 0, the hypothesis of “exact-fit” (i.e. the null hypothesis that RMSEA in the population is 0) is retained (α = 0.05). In addition, the less stringent “close-fit hypothesis” (i.e. the null hypothesis that RMSEA in the population is less than or equal to .05) is tested. A p-value (denoted as pclose) of this test which is greater than α (i.e. statistically insignificant) supports the model.

4. Page 13/line 13: (Pearson) bivariate product moment correlations are presented in Table 2. In the footnote of Table 2 (Page 15), statistically significant correlation coefficients are highlighted. As data do not follow a multivariate normal distribution (as mentioned at page 11/line 8), what is the adopted procedure to test the correlations?

In Table 2 standard parametric t-tests were employed. We cross-checked the results of the significance test using bootstrapping (percentile method, 1000 samples). All 110 statistical decisions (statistically significant or not) derived from the parametric test and the bootstrapping procedure were identical except r(FKS, MGHHPS) in the full sample which became statistically significant only using the parametric method. Nevertheless, we now report the results of the bootstrapping tests instead of the parametric test in Table 2 which is introduced on page 13:

“Since data were not bivariate normally distributed we employed bootstrapping instead of the standard parametric significance tests for r (against 0, percentile method, 1000 samples)”.

Reviewer #2: PONE-D-19-15829 

It is a very interesting work, very well justified, analyzed and explained. The sample used is large, the instruments used are relevant, the statistical applied are flawless, and the contributions are adequately discussed, including the limitations of the study.

Thank you very much for this positive evaluation of our manuscript. We have answered your important questions and concerns in the following.

The researchers are asked some questions that we consider important from a conceptual point of view. This has to be taken into account, since it is the main direction and contribution of this research.

5. First of all, one should not speak of BDD but of BDD symptomatology. This should be taken into account because some questionnaires or scales have been administered, but it cannot be confirmed in this way that we are talking about a formal diagnosis. The authors allude precisely to the DSM and ICD, so it is understood that we should talk about a finding of inclusion, exclusion, and impairment criteria identified through a formal diagnostic interview. In this way we could say, strictly speaking, that we are talking about BDD.

Thank you very much for this important remark. Indeed, we aimed to relate BDD psychopathology (but not the disorder as an entity) with other psychopathologies and agree with the Reviewer, that we did not formally diagnose our participants. We have, therefore, rephrased throughout the manuscript as “BDD symptoms”. 

6. In relation to the above, the authors are suggested to refer to symptoms of BDD, or sub-threshold BDD or, as expressly referred to by the DSM, dysmorphic concerns. In the same vein, authors are recommended to review a publication that addresses precisely this issue (doi: 10.1002 / ijop.12646).

We are thankful for the comment and the recommendation of this interesting publication. As we have employed the Body Dysmorphic Symptoms Inventory (German: Fragebogen Körperdysmorpher Symptome) which assesses not only dysmorphic concerns but also related behaviors we decided to rephrase as “BDD symptoms” throughout the manuscript.

7. Secondly, being in full agreement with the authors about the approach adopted, and the results indicated, it is perhaps missing what aspects of body image are relevant here. It may seem tautological that there are disorders where the body and discomfort about it are categorized more precisely as body image disorders. It would be relevant to propose what aspects should lead the investigation if, as proposed, the validity of the construct has more to do with body image than with emotional, obsessive-compulsive symptoms, etc.

Thank you for this chance to clarify the concept of body image and body image disturbance more thoroughly for the readers as well as point out more specifically which aspects seem relevant for a distinct classification in a body image spectrum vs. an emotional, ocd spectrum etc. We follow the definitions of Cash (2004) and Slade (1988) that summarize body image to consist of four components, cognitive, perceptive, affective and behavioral. Thus, disorders that might be categorized in this category are not only characterized by a discomfort about the own body but also about misperceptions, behaviors like checking and avoidance as well as cognitive distortions such as attention or interpretation biases. Thus, symptoms comparable to those in emotional disorder or OCRD might be present but are all related to the body (sadness about the body, compulsive checking the body, etc). We have made this clearer in the discussion section on pages 22-23):

“Disorders falling in such a category would be characterized by a body image disturbance, involving an affective component such as body dissatisfaction, a misperception of the body, cognitive distortions such as attention and interpretation biases, and behaviors such as checking, avoidance, and appearance fixing [80, 81]. Thus, these disorders might present with symptoms also present in other disorders such as sadness or OCD-like checking, but they are all exclusively related to the body.”

8. Third, in connection with the above, it is true, as the authors say, that there is no research about the relationship between somatization and BDD. It may be necessary to point out components of somatization that must be taken into account in order to really discard this relationship. Somatization cannot be analyzed simply from a list of physical complaints. The description of the absorption states that can be seen in both the OCD and the BDD is classic. Precisely for that reason, it is a less analyzed, but proposed psychopathological organizer. It is suggested to review: doi: 10.1016 / j.cpr.2004.08.006 as well as Farina, B., & Liotti, G. (2013). Does a dissociative psychopathological dimension exist? A review on dissociative processes and symptoms in developmental trauma spectrum disorders. Clinical Neuropsychiatry, 10(1), 11–18.

Thank you very much for the recommendation of these interesting reads. Indeed, this is a potential psychopathological organizer that might link the disorders. We have provided this information now in our introduction, also in order with Reviewer #3’ suggestion (see comment #14) to strengthen the empirical basis of our models. See page 4 of the manuscript:

“On the other hand, several somatoform disorders and BDD [17, 18] amongst other disorders share a heightened likelihood for dissociation and thus might be linked via dissociation as a psychopathological organizer [19, 20]. 

9. In short, the authors are asked to indicate which psychopathological organizers would conform to the BDD as a body image disorder, close, but differentiable from the Eating Disorders, for example. This contribution could lead the investigation precisely on the mediating and moderating components, and not exclusively on the description of symptoms and behaviors.

We agree with the Reviewer that psychopathological organizers are highly relevant, in particular when following a non-categorical but rather dimensional approach, e.g., as suggested within the Research Domain Criteria (RDoC) approach by the National Institute for Mental Health (NIHM) (Cuthbert, 2015). In line with the Reviewer’s suggestion, we believe that is important to mention this alternative approach to classification of mental disorders and highlight it now in the discussion section on pages 24-25: 

“Despite this highly important finding, it is of note, that besides this rather categorical approach of classification, recently, a dimensional one has been proposed and highly researched. Along the lines of the initiative Research Domain Criteria (RDoC) of the National Institute of Mental Health (e.g., [89]), researchers aim to organize mental disorders not only based on symptoms and behaviors presented but rather along domains of human functioning such as negative valence or cognition that can be measured rather diversely (e.g., self-report vs. on a physiological or a genetic level). Thus, future research might also want to be focused on finding common psychopathological organizers of BDD with other disorders and examine where BDD lies on these dimensions measured with different instruments, respectively.”

Furthermore, we have added a recent study that has highlighted that EDs (and in particular anorexia nervosa) and BDD are most easily differentiated by the body part(s) individuals are concerned with (see page 23): 

“Disorders within this category might be most easily differentiated by examining the body parts individuals are most dissatisfied with, i.e., body zones typically linked to weight concerns in EDs, particularly anorexia nervosa, whereas facial features, hair and skin in BDD [82].”

Reviewer #3: 

The current manuscript employs structural equation modeling (SEM) to identify an empirically supported classification structure for body dysmorphic disorder (BDD). The authors generated six a priori models based on the literature and determined that a Body Image Model provided the best fit structure. This is an important and understudied area of research. Particular strengths of this study, relative to past research on this topic, are the use of SEM with strong theoretically-derived models, large sample size, and a set of well-validated indicator measures. However, there are some concerns with the analyses and suggestions described in more detail below to improve interpretability of the findings.

Thank you very much for your favorable evaluation of our manuscript and your critical comments. We addressed each of the latter below.

10. The authors did a commendable job in their description, justification, and interpretation of all fit indices, and examination of gender invariance and multi-group solutions. However, the main concern with the approach is with variable selection.

 There would be an expected level of multicollinearity among indicator variables across the spectrum of psychopathology included in this study, but there is quite substantial collinearity between the FKS and EDE-Q, in particular, at the zero-order level (r = .798), which is only potentially problematic in that suggests there may be redundant information contributing to the “Body Image” factor. Given that this model provided the best fit, I wonder if the authors inspected reasons for this particularly high degree of overlap and considered dropping each of the variables (e.g., EDE-Q) to test its effect on parameter estimates and model fit. Were there other eating disorder inventories/scales included in the online survey? Alternatively, I wonder if there are subscales within the FKS and EDE-Q that particularly hang together driving the overall multicollinearity, which could be examined in further sensitivity analyses to defend the robustness of the Body Image model.

This is an important remark. Unfortunately, in order to keep the survey as short as possible, we did not have other eating disorder measures. 

To test for the robustness of the results we followed your idea and omitted each of the three indicators FKS, EDE-Q, and MDDI successively from the body-image factor in model 5. These following SEM results were obtained (in the following always using the complete sample):

SEM Fit Statistics for models 5 (BIM) with two indicators for the body image factor

Model SB-�2 df CFI TLI RMSEA [90% CI] pclose SRMR AIC BIC

complete 104.096*** 38 0.972 0.959 0.049 [0.039, 0.059] .57 0.027 12,007.593 12,136.427

without 

FKS 87.169*** 29 0.966 0.948 0.052 [0.041, 0.063] .35 0.028 11,319.933 11,439.565

without EDE-Q 86.668*** 29 0.967 0.949 0.052 [0.041, 0.063] .37 0.027 10.214.924 10,334.556

without MDDI 88.550*** 29 0.969 0.952 0.053 [0.042, 0.064] .32 0.028 11,200.803 11,320.435

It can be seen that all three models (a) show an acceptable to good global fit, (b) which is slightly lower compared to the complete, more complex model and (c) do not differ much among each other with respect to fit. This shows that dropping one of the three indicators do not change the results significantly.

In a second step we took the worst fitting model (the one without FKS) and compared all six models omitting always the indicator FKS. The following results show that model 5 is again clearly superior to any other model:

SEM Fit Statistics for all Models without indicator FKS

Model SB-�2 df CFI TLI RMSEA [90% CI] pclose SRMR AIC BIC

1 252.541*** 35 0.875 0.839 0.092 [0.083,0.101] .00 0.053 11,540.394 11,632.419

2 250.697*** 32 0.874 0.823 0.096 [0.087, 0.106] .00 0.051 11,535.138 11,640.966

3 240.262*** 32 0.880 0.831 0.094 [0.085, 0.104] .00 0.052 11,532.415 11,638.244

4 247.242*** 32 0.876 0.826 0.110 [0.097, 0.123] .00 0.050 11,527.104 11,632.932

5 87.169*** 29 0.966 0.948 0.052 [0.041, 0.063] .35 0.028 11,319.933 11,439.565

6 237.418*** 30 0.880 0.821 0.097 [0.087, 0.107] .00 0.049 11,552.218 11,637.249

Together these results speak in favor of the robustness of the results.

A second interesting question is whether the multicollinearity is due especially to one of the subscales of the EDE-Q (there are no subscales for FKS and MDDI). To further investigate this question, we successively left out all items which belong to one of the four subscales (restraint, eating concern, weight concern, shape concern) from the total EDE-Q score. The four reduced total scores (comprised of the items from three out of four subscales) correlate highly among each other (all r’s > .95). Furthermore, they show similar correlations with the other two body image indicator FKS and MDDI, which can be seen here:

 Items omitted from EDE-Q subscale 

 restraint eating concern weight concern shape concern complete

FKS .811 .788 .796 .756 .798

MDDI .692 .683 .686 .638 .684

Taking the reduced total scores as indicators results in the following global fit statistics of SEM:

SEM Fit Statistics for Models 5 (BIM) with reduced total score indicators for EDE-Q.

Model 5 SB-�2 df CFI TLI RMSEA [90% CI] pclose SRMR AIC BIC

complete 104.096*** 38 0.972 0.959 0.049 [0.039, 0.059] .57 0.027 12,007.593 12,136.427

without 

restraint 101.176*** 38 0.974 0.962 0.048 [0.038, 0.058] .64 0.027 12,023.215 12,152.049

without eating concern 106.658*** 38 0.971 0.958 0.050 [0.040, 0.060] .51 0.028 12,154.677 12,283.511

without weight concern 105.050*** 38 0.971 0.959 0.049 [0.039, 0.059] .55 0.027 11,985.821 12,114.655

without shape concern 102.974*** 38 0.971 0.958 0.048 [0.038, 0.058] .60 0.028 12,064.356 12,193.190

Here, again, all four models differ very little among each other with respect to fit. 

Together, the additional analyses show that there is no unique EDE-Q subscale driving the overall multicollinearity. Furthermore, this provides additional evidence for the robustness of the results in favor of the Body Image model.

Given the constraints on overall length of the manuscript, we would not add these additional analyses to the manuscript, unless the Reviewer and/or Editor believe them to be indispensable.

12. There were few indicator variables (as few as 2) for some of the factors. Please discuss the implications of this on parameter estimation and model comparison.

Thank you very much for bringing this to our attention. We have now provided a discussion on page 24 of the manuscript: 

“Also, as can be seen from Figure 1, some factors were assessed only by few indicator variables. In general, simulations show that more indicators per factor result in fewer nonconverged solutions, fewer improper solutions and more accurate parameter estimates (MBR; [88]). But simulations demonstrate as well that a low number of indicators can be compensated by larger sample size, e.g. N ≥ 400 [61]. Moreover, there were no signs of nonconvergent or improper solutions in any of our models. Taken together, we found no evidence that the relatively low number of indicators has caused problems. “

13. Another concern is regarding the significant amount of missing data, which is common to survey studies. How were “complete” surveys defined, and were there any attempts made to examine patterns of missingness based on available data collected?

Indeed, the amount of missing data is negligible. None of the symptom scales had more than three missing values (<0.03%). In total there were only 34 missing values across all items and scales (<0.03%). Given the small numbers of missing values no particular pattern could be detected.

However, we have included only the survey answers of individuals that have completed all questionnaires that are part of this study. Two questionnaires at the end of the survey were not part of this main study. 

We have added respective information in the manuscript on pages 7 and 13: 

“(i.e., filled out all questionnaires included in this main manuscript results of two further questionnaires will be reported elsewhere);“

“None of the symptom scales had more than three missing values (<0.03%), with only 34 missing values across all items and scales in total (<0.03%).”

14. It was clear to me that the six models tested in the current study were derived based on the existing literature; however, to an unfamiliar reader, this may have been unclear. There could be more background on the selection of measures and models included in the study. I also couldn’t find figure captions for Figures 1 and 2.

Thank you very much for this comment. In line with Reviewer 1 comment # 8, we have added a bit more empirical information as to why we test the somatoform model as one option. We believe that we have provided the adequate empirical backgrounds including sources for the other potential models. To underline the fact that the potential models are based on empirical or theoretical (past or current classification) data, we have made that clearer right before providing our research question on pages 6-7: 

“Based on the reviewed literature, mainly of transdiagnostic studies comparing BDD mostly with OCD, eating disorders, anxiety, and depression as well as the study by Schneider and colleagues [39] analyzing psychopathological data in adolescents, but also past and current classification of BDD, six potential classification models were specified”

Furthermore, we shortly summarized that these German instruments selected were chosen based on brevity, representativeness of respective symptomatology, as well as frequency of their use. Due to space constraints we do not provide explanations separately for each questionnaire. See page 8:

“Instruments were chosen based on their brevity, representativeness of symptomatology, and frequency of use.”

 Additionally, we have included the Figure Captions at the end of the manuscript now (see page 32).

15. Results on clinical characteristics appear to be missing on p.13. 

Thank you very much for catching this mistake. As we have decided to present means and standard deviations of the questionnaire scores in a table, we abstained from presenting it in this section again, but forgot to delete this part of the title, which we now did.

16. As the study objective was to examine the structure of BDD, it presents a challenge interpreting data based on a community (non-clinical) sample. The defining feature of BDD is that the physical features are dysmorphic, by nature, which cannot be reliably assessed via self-report. Although the authors already note these limitations in the Discussion, some of the findings are still overstated and I suggest scaling back the language accordingly. In particular, I suggest paying close attention to discussion of the findings with reference to studies conducted in large community samples that estimate the whole continuum of body dysmorphic concerns (and other related pathology). For example, the authors discuss the implications of support for a body image spectrum of classification for BDD but only in the context of clinical studies of BDD, eating disorders, and MD (p.21), rather than in non-clinical, analogue, or community samples.

Thank you very much for this important comment. We strengthened the limitation point accordingly on page 24:

“Along the same lines, we have only examined BDD symptoms independent of those warranting a diagnosis of BDD. Therefore, it might be advisable to replicate these findings in a clinical sample with subthreshold levels of symptoms looking into the comorbidity structure at the disorder level.”

Furthermore, we have included another publication (besides the Schneider et al. [2018] study) that has looked at dysmorphic concerns and ED symptoms in a non-clinical sample that supports our data (see page 22): 

“Regarding non-clinical samples, Samad and colleagues [77] found that while ED and BDD symptoms seem to track together, they seem to identify different sets of psychopathological features. This supports our results that the two symptomatologies might best be captured under one category.“

Also, in line with comment #5 by Reviewer #1 we have rephrased BDD throughout the manuscript to BDD symptoms when speaking about our sample, as we did not formally diagnose our participants. 

And lastly, we have formulated an implication aspect in the last paragraph that focuses on non-clinical samples (page 25): 

“With regard to non-clinical samples, the high correlation of symptoms in the present sample highlights the need to address the potentially present other symptom cluster in prevention programs.”

---

## [Decision Letter · Decision Letter 1]

30 Apr 2020

An empirically derived recommendation for the classification of body dysmorphic disorder: Findings from structural equation modeling

PONE-D-19-15829R1

Dear Dr. Hartmann,

We are pleased to inform you that your manuscript has been judged scientifically suitable for publication and will be formally accepted for publication once it complies with all outstanding technical requirements.

With kind regards,

Flávia L. Osório, PhD

Academic Editor

PLOS ONE

Additional Editor Comments (optional):

The reviewers considered that the authors responded adequately to all questions. The article is ready to be published

Reviewers' comments:

Reviewer's Responses to Questions

**Comments to the Author**

1. If the authors have adequately addressed your comments raised in a previous round of review and you feel that this manuscript is now acceptable for publication, you may indicate that here to bypass the “Comments to the Author” section, enter your conflict of interest statement in the “Confidential to Editor” section, and submit your "Accept" recommendation.

Reviewer #1: All comments have been addressed

Reviewer #2: All comments have been addressed

Reviewer #3: All comments have been addressed

2. Is the manuscript technically sound, and do the data support the conclusions?

Reviewer #1: Yes

Reviewer #2: Yes

Reviewer #3: Yes

3. Has the statistical analysis been performed appropriately and rigorously? 

Reviewer #1: Yes

Reviewer #2: Yes

Reviewer #3: Yes

4. Have the authors made all data underlying the findings in their manuscript fully available?

Reviewer #1: Yes

Reviewer #2: Yes

Reviewer #3: Yes

5. Is the manuscript presented in an intelligible fashion and written in standard English?

Reviewer #1: Yes

Reviewer #2: Yes

Reviewer #3: Yes

6. Review Comments to the Author

Reviewer #1: The article is about a relevant topic, it is well-written and a pleasant reading. It considers appropriate

statistical methodologies, which are well described. The authors answered all previous questions satisfactorily. I have no additional comments.

Reviewer #2: Congratulations to the authors for the efforts made to clarify differents aspects from the review in the article. I can see the authors have worked hard to improve the research comprehension for the readers and I would like to express my appreciation for these efforts made. The paper is now ready for publication, and I wish the authors all the best with their further research.

Reviewer #3: The authors have done a commendable job addressing all of my concerns and I agree that the additional analyses not only support the robustness of the best fit model but also do not need to be reported in the manuscript.

7. PLOS authors have the option to publish the peer review history of their article (what does this mean?). If published, this will include your full peer review and any attached files.

Reviewer #1: No

Reviewer #2: Yes: Juan F. Rodriguez-Testal

Reviewer #3: No

---

## [Editor Report · Acceptance letter]

20 May 2020

PONE-D-19-15829R1 

An empirically derived recommendation for the classification of body dysmorphic disorder: Findings from structural equation modeling 

Dear Dr. Hartmann:

I am pleased to inform you that your manuscript has been deemed suitable for publication in PLOS ONE. Congratulations! Your manuscript is now with our production department. 

With kind regards,

on behalf of

Dr. Flávia L. Osório 

Academic Editor

PLOS ONE